# Hepatitis E and Allogeneic Hematopoietic Stem Cell Transplantation: A French Nationwide SFGM-TC Retrospective Study

**DOI:** 10.3390/v11070622

**Published:** 2019-07-05

**Authors:** Aliénor Xhaard, Anne-Marie Roque-Afonso, Vincent Mallet, Patricia Ribaud, Stéphanie Nguyen-Quoc, Pierre-Simon Rohrlich, Reza Tabrizi, Johanna Konopacki, Séverine Lissandre, Florence Abravanel, Régis Peffault de Latour, Anne Huynh

**Affiliations:** 1Service d’hématologie-greffe, Hôpital Saint-Louis, Université Paris-Diderot, 75010 Paris, France; 2Service de virologie, Hôpital Paul-Brousse, 94804 Villejuif, France; 3INSERM 1193 et CNR hépatite A et E, Université Paris-Sud, 94804 Villejuif, France; 4Service d’hépatologie, Hôpital Cochin, Université Paris Descartes, INSERM U1223, Institut Pasteur, 75014 Paris, France; 5Service d’hématologie, Hôpital La Pitié-Salpêtrière, 75013 Paris, France; 6Service d’hématologie, Centre Hospitalier Universitaire, 06000 Nice, France; 7Service d’hématologie, CHU Bordeaux, 33600 Pessac, France; 8Service d’hématologie, Hôpital Percy, 92140 Clamart, France; 9Service d’hématologie, CHU, 37000 Tours, France; 10Laboratoire de virologie et CNR hépatite E, CHU, 31000 Toulouse, France; 11Service d’hématologie, CHU, 31000 Toulouse, France

**Keywords:** allogeneic hematopoietic stem transplantation, hepatitis E

## Abstract

Usually self-limited, hepatitis E virus (HEV) infection may evolve to chronicity and cirrhosis in immunosuppressed patients. HEV infection has been described in solid-organ transplantation and hematology patients, but for allogeneic hematopoietic stem cell transplant (alloHSCT) recipients, only small cohorts are available. This retrospective nationwide multi-center series aimed to describe HEV diagnostic practices in alloHSCT French centers, and the course of infection in the context of alloHSCT. Twenty-nine out of 37 centers participated. HEV search in case of liver function tests (LFT) abnormalities was never performed in 24% of centers, occasionally in 55%, and systematically in 21%. Twenty-five cases of active HEV infection were diagnosed in seven centers, all because of LFT abnormalities, by blood nucleic acid testing. HEV infection was diagnosed in three patients before alloHSCT; HEV infection did not influence transplantation planning, and resolved spontaneously before or after alloHSCT. Twenty-two patients were diagnosed a median of 283 days after alloHSCT. Nine patients (41%) had spontaneous viral clearance, mostly after immunosuppressive treatment decrease. Thirteen patients (59%) received ribavirin, with sustained viral clearance in 11/12 evaluable patients. We observed three HEV recurrences but no HEV-related death or liver failure, nor evolution to cirrhosis.

## 1. Introduction

Hepatitis E virus (HEV) is endemic worldwide and a leading cause of waterborne acute hepatitis in developing countries, mostly due to genotypes 1 and 2. In many European countries, genotype 3 is prevalent and locally acquired HEV has become the most common cause of acute viral hepatitis [1]. The infection is mostly of zoonotic origin, usually through contaminated food (mainly undercooked pig meat). HEV infection can also be contracted via contaminated blood products, either by transfusion or via stem cell products in cases of allogeneic hematopoietic stem cell transplantation (alloHSCT) [2,3,4,5,6,7,8]. In France, HEV seroprevalence increases with age, from 4.7% in the 2–5 year age group to 70% in the 58–65 year age group, suggesting a lifelong exposure to HEV [9]. HEV infection is usually self-limited, but may evolve to chronic hepatitis E (defined by the persistence of HEV replication for six months) [10] and cirrhosis in immunosuppressed patients [10,11,12]. Patients with chronic hepatitis E usually clear HEV when immunity is restored.

While knowledge and awareness on HEV infection is increasing [13,14], especially in immunosuppressed hosts, HEV infection is frequently undiagnosed, clinical practices remain heterogeneous, and fatal cases have been reported [10,15,16]. We conducted a retrospective French nationwide study with two aims: firstly, to describe HEV diagnostic practices in alloHSCT centers, and secondly to describe the course of HEV infection before or after alloHSCT.

## 2. Materials and Methods

All adult and pediatric alloHSCT centers (*n* = 37) members of the SFGM-TC (Société Francophone de Greffe de Moelle et de Thérapie Cellulaire) were invited to participate in this retrospective survey. Questionnaires sent in 2014 and 2015 requested the number of acute or chronic HEV infections diagnosed before or after alloHSCT, circumstances in which HEV NAT (nucleic acid testing) or serology were performed, nature of samples (blood or stools) used for NAT and the date of the first HEV search in each center. Data were collected from patient files and virology laboratories. Since the design of the study was multicentric and retrospective, the assays used for NAT and serology varied over time and differed according to each center’s laboratory. Both commercially available kits and in-house RT-PCR techniques were used for NAT detection and genotyping. Commercially available kits were used for IgG and IgM detection. A case of infection was defined as a patient with positive HEV NAT in serum or plasma. The study was conducted according to the Helsinki declaration. All participants gave written informed consent for anonymous data collection that was declared to the appropriate authorities.

## 3. Results

Twenty-nine centers participated: 25 cases were diagnosed in seven centers (range per center: 1–7) (Figure 1). Three cases were diagnosed before alloHSCT between 2005 and 2009 in the center that has worked the most on HEV infection in alloHSCT and solid-organ transplantation (SOT) recipients. All other cases were diagnosed after alloHSCT between 2010 and 2015.

### 3.1. HEV Diagnostic Procedures

Seven centers (24%), including pediatric units, never searched for HEV in cases of increased transaminases. Sixteen centers (55%) occasionally searched for HEV if transaminases increased; five cases were diagnosed in two centers. Six centers (21%) systematically looked for HEV when transaminases increased; 20 cases were diagnosed in five centers, including three cases before alloHSCT.

### 3.2. Patient Characteristics at HEV Infection Diagnosis

Patient and transplant characteristics are presented in Table 1. Eighteen patients (72%) were male, and the median age was 49 years (range: 23–68). HEV infection was diagnosed by detectable blood HEV NAT, because of liver function test (LFT) abnormalities, before alloHSCT in three patients and after alloHSCT in 22. Three centers, corresponding to eight patients, also investigated stool samples. At diagnosis, NAT was positive in stools in seven patients. Three patients died four months, 18 months, and 4.5 years after alloHSCT, respectively, due to bacterial infection, fungal infection or relapse, respectively. Two patients died after viral clearance, and one patient died with decreasing but still positive HEV NAT. No death was attributed to HEV infection.

### 3.3. HEV Cases Diagnosed before Transplantation

All three patients had received chemotherapy, none had neutropenia and one (33%) had lymphopenia. All three patients received an alloHSCT from a matched related donor after a myelo-ablative conditioning regimen (Table 1). None were treated for HEV infection.

The follow-up of LFT abnormalities, NAT and serology is presented in Appendix A. Genotyping of viral strains was not performed. In one patient, HEV infection was diagnosed eight days before alloHSCT, at the beginning of conditioning. Conditioning and transplantation were not delayed. HEV NAT was positive in blood and stools until five months after alloHSCT. In the two other patients, HEV infection was diagnosed 43 and 88 days before alloHSCT, with spontaneous viral clearance before alloHSCT. Blood HEV NAT remained negative during all follow-up for one patient and became transiently detectable two months after alloHSCT for the other one.

### 3.4. HEV Cases Diagnosed after Transplantation

Twenty-two cases were diagnosed a median of 283 days (range: 26–4177) after alloHSCT (Table 1). All patients had elevated transaminases. The median increases in LFT were AST 3.2 × ULN (range: 1–48), ALT: 6.2 × ULN (range: 1–68), GGT: 5.9 × ULN (range: 1.5–17.5), ALP: 1.2 × ULN (range: 1–4.5), and bilirubin: 1–27 µmol/L. At diagnosis, no patients were neutropenic. The lymphocyte count was <1000/mm^3^ for nine patients (no immunophenotyping available). Eighteen patients were receiving immunosuppressive treatment (steroids 0.1–0.5 mg/kg/day and/or calcineurin inhibitor and/or mycophenolate mofetil). Two patients had received chemotherapy for relapse after alloHSCT. The HEV genotype was known for six patients: 3c (*n* = 2) or 3f (*n* = 4). Liver biopsies were performed in four patients with positive HEV NAT at the time of biopsy (Table 2). Significant fibrosis was revealed in a patient with a 14-day history of HEV infection, who acknowledged chronic alcohol consumption. One patient had two biopsies, performed during HEV infection but before diagnosis. After HEV diagnosis and retrospective NAT of frozen plasma samples, it appeared that biopsies were performed after at least one and four months of HEV infection. Regenerating activity was observed in both biopsies, without fibrosis. No fibrosis was observed in the other two patients, who had a six-month history of HEV infection.

In three (14%) patients (Patients 13, 14 and 24), an HEV infection preceding alloHSCT was diagnosed after alloHSCT (two, seven and 12 months, respectively) by retrospective HEV NAT of pre-alloHSCT frozen serum or plasma samples.

HEV serology was available for 17 patients (Table 3). Serology was performed at diagnosis for 12 patients, with no reliable pattern.

### 3.5. Treatment of HEV Infection

The decision to start ribavirin treatment was taken on a per-patient basis, depending on immunosuppressive treatment, time from alloHSCT, and physician experience. HEV infection was diagnosed a median of 280 days after alloHSCT (range: 78–4177) in patients not treated with ribavirin and a median of 287 days after alloHSCT (range: 26–1926) in patients treated with ribavirin. At HEV diagnosis, the percentage of patients who did not receive immunosuppressive treatment was the same (22%) in patients treated with ribavirin or not.

Nine patients did not receive any anti-HEV treatment, and spontaneous viral clearance occurred one to nine months after diagnosis (Appendix A). Seven patients were receiving immunosuppressive treatment, which was decreased. LFT values were increased at diagnosis and improved over time for all patients. LFT normalization could be observed as soon as two weeks from HEV diagnosis and could be seen before blood NAT was found negative. Conversely, LFT could still be abnormal in patients with negative blood NAT, as LFT abnormalities may be multi-factorial in alloHSCT patients.

Thirteen patients were treated with ribavirin (400–1000 mg/day), begun a median of 18 days after HEV diagnosis (range: 1–229). The median treatment length was 90 days (range: 24–345). The treatment course with NAT follow-up is presented in Appendix A. During ribavirin treatment, three patients had cytopenias (pancytopenia, neutropenia, erythroblastopenia, *n* = 1, each), leading to treatment withdrawal. LFT normalized within six weeks after treatment initiation for 10 patients but remained increased in three patients despite negative NAT. HEV blood NAT turned undetectable in 12 evaluable patients a median of 2.3 months (range: 1–9) after ribavirin initiation. Patient 14 died two months after beginning treatment, with decreasing but still positive NAT. Death was due to fungal infection in the context of secondary leukemia. All 12 patients had negative HEV blood NAT at the end of treatment and 11 had sustained viral clearance with undetectable HEV NAT three months after ribavirin was discontinued. For Patient 13, HEV blood NAT was negative at the end of ribavirin treatment and positive again two months later, with spontaneous clearance within five months. Patient 22 presented a specific HEV infection course: HEV infection was diagnosed 83 days after alloHSCT, with spontaneous viral clearance within three months. The patient then received donor lymphocytes injection for mixed chimerism, and developed graft-versus-host-disease, treated with triple immunosuppression. One month later, HEV blood NAT was positive again, with increased LFT. The patient was successfully treated with ribavirin, with improved LFT and viral clearance (Appendix A).

## 4. Discussion

This is the first study to report HEV infection in alloHSCT patients at a nationwide level. HEV infection is probably under-diagnosed, as one quarter of French alloHSCT centers never look for HEV, and more than half of the centers do not systematically perform HEV NAT in cases of LFT abnormalities. Twenty-five cases are reported in this study. Half of the cases were diagnosed in the south of France, where the seroprevalence is the highest [17,18] and where physicians are more aware of the infection. However, 11 cases were diagnosed in three centers of Paris and its region, suggesting that HEV is more frequently diagnosed when the physician’s awareness of HEV is high. Alongside physician awareness of HEV, the different techniques used for HEV NAT among centers might have played a role in the diagnostic performance of each center. Treatment decision was at the attending physician’s discretion. For three cases diagnosed before alloHSCT, viral clearance occurred spontaneously before (*n* = 2) or after (*n* = 1) alloHSCT. Among the 22 patients diagnosed after alloHSCT, nine cleared HEV infection within one to six months, with no specific intervention in patients who were not receiving immunosuppressive drugs (*n* = 2) or with immunosuppressive treatment reduction (*n* = 7). Thirteen patients were treated with ribavirin, with sustained viral clearance in 11 out of 12 evaluable patients.

HEV infection is easily overlooked. In our population, HEV infection was mostly asymptomatic (76%), whereas up to 35% of symptomatic patients have been reported among SOT recipients [17]. In alloHSCT patients, LFT abnormalities are commonly incorrectly ascribed to drug-induced liver injury or graft-versus-host disease. Although a useful tool for HEV infection diagnosis in the general non-immunosuppressed population, serology was not reliable for HEV diagnosis or follow-up in our population: serology was not always positive at diagnosis, we observed loss of anti-HEV IgG during follow-up and some patients cleared HEV without seroconversion. Aberrant serodynamics have previously been reported among alloHSCT recipients [15]. Impaired immune reconstitution and hypogammaglobulinemia are frequent after alloHSCT and can explain these findings. Moreover, IgG could be acquired via the infused graft or blood products and do not protect from a new infection [18,19,20]. HEV infection diagnosis rests on the demonstration of HEV RNA by NAT and should be included in the diagnostic work-up of LFT abnormalities in alloHSCT recipients.

In all our cases, the origin of the infection remained unknown but was difficult to assess due to the retrospective nature of the study. Because of their immunosuppressed state, alloHSCT recipients are usually advised not to eat raw meat, but food contamination cannot be formally excluded, especially in cases where contamination occurred several months to years after alloHSCT, when dietary rules are less stringent. French blood donors are not systematically screened for HEV infection, and no HEV-free packed red blood cells or platelets are available. In six patients from the same center, stem cell graft and blood products transfused in the three months preceding HEV infection diagnosis were explored, but no case of HEV transmission via infused blood products or stem cell graft could be found. (AX, [21]).

All the nine patients that were not treated for HEV infection had spontaneous viral clearance within one to six months after HEV diagnosis. This contrasts with the 66% evolution rate to chronic hepatitis E among 85 SOT recipients [17]. The differences in intensity and length of immunosuppressive treatment between alloHSCT and SOT recipients probably partly explain this difference in the spontaneous evolution of HEV infection. Versluis et al. previously reported a 60% rate of evolution towards chronic hepatitis E in eight alloHSCT recipients [15]. The median time from alloHSCT to HEV diagnosis in that population was 4.6 months, versus 9.4 in our study. Differences in the immunosuppressive status of both populations probably explain this difference. Previous reports have correlated the level of immunosuppression with the outcome of HEV infection [19]. For our patients diagnosed after alloHSCT and not treated for HEV infection, immunosuppressive treatment was decreased, probably contributing to a partial restoration of immunity and allowing viral clearance. Indeed, for SOT recipients, a reduction of immunosuppression is recommended as a first-line therapeutic option, as it can increase T-cell response and allow HEV clearance in up to 30% of cases [22,23].

A review of HEV infection in alloHSCT recipients published in 2014 [14] and recently published ECIL-5 European Conference on Infections in Leukaemia [22] and EASL European Association for the Study of the Liver [1] guidelines for HEV also recommend an immunosuppressive treatment decrease for patients with hematological malignancies and chronic hepatitis E. When it is impossible to decrease immunosuppressive treatment, or in cases of failure to clear HEV after immunosuppressive treatment decrease, ribavirin treatment is recommended [22]. Most patients are treated on a per-patient basis, depending on immunosuppressive treatment at diagnosis and time from alloHSCT.

Sustained viral clearance occurred with ribavirin treatment in 11 out of 12 evaluable patients. Compared with the SOT recipients population, treatment was initiated earlier after HEV diagnosis (median time: 18 days versus nine months), but the median treatment length was the same (90 days) [23]. Viral clearance with ribavirin treatment could be observed as soon as 30 days after treatment was initiated. Three patients presented with cytopenias during ribavirin treatment, leading to treatment withdrawal in all cases. Hemoglobin levels, transfusions, erythropoietin administration and ribavirin plasma levels were not assessed in this retrospective study. In SOT recipients, more than half of the patients treated with ribavirin received recombinant erythropoietin treatment [9]. Hematological tolerance of ribavirin treatment should be studied in a future prospective study, especially in a population of alloHSCT recipients, who are prone to cytopenias.

In three cases, we observed a new episode of HEV NAT positivity after a period of negativity. For these three cases, one, two and three blood NATs were negative, respectively (corresponding to a one to three-month period). One patient diagnosed before alloHSCT had spontaneous viral clearance before alloHSCT and transient positive blood NAT two months after alloHSCT that became spontaneously negative. Another patient diagnosed after alloHSCT spontaneously cleared HEV but was found to be positive again after immunosuppressive treatment intensification with successful ribavirin treatment. One patient with a negative blood NAT at the end of ribavirin treatment had a positive HEV NAT two months after ribavirin withdrawal, and became spontaneously negative within five months. Abravanel et al. [24] studied 36 HSCT recipients (including 29 allogeneic transplantations) with a pre-transplantation positive HEV serology (IgG with or without IgM): HEV NAT remained negative for all patients up to six months after transplantation. On the contrary, two publications have reported two cases of HEV recurrence after alloHSCT [15,25]: both patients had a pre-transplant acute HEV infection and post-alloHSCT new episode of NAT positivity after a period of NAT negativity. In both cases, sequenced strains were identical during the two episodes. Unfortunately, as our study was retrospective, most HEV strains were not genotyped. HEV recurrence only a few weeks or months after viral clearance, in the context of a recent alloHSCT in a patient receiving an immunosuppressive treatment, could be explained by an incomplete viral clearance by the immune system and/or by the persistence of the virus in tissues [25]. Another hypothesis could be the persistence in the blood of very low concentrations of HEV RNA, below the detection limit of the NAT, leading to a false-negative NAT [24]. A new episode of NAT positivity could also be related to a new infection by a different HEV strain [18,19,20]. In the SOT recipient population successfully treated with ribavirin, an 18% rate of HEV NAT positivity after ribavirin treatment withdrawal has been observed [23]. Once again, the differences in immunosuppressive treatment length and intensity between alloHSCT and SOT recipients might explain this difference.

The evolution of chronic hepatitis E towards cirrhosis and its complications, including decompensated cirrhosis and hepatocellular carcinoma, has been reported in HIV-positive patients and SOT and alloHSCT recipients [10,11,15,25]. No such evolution to cirrhosis was observed in our population, and even liver biopsy performed in our two patients with a six-month history of NAT positivity did not show any signs of liver fibrosis. In one patient, liver biopsy showed significant inflammation, with chronic hepatitis and signs of cirrhosis, but the biopsy was performed only two weeks after HEV diagnosis and the patient acknowledged chronic alcohol consumption.

HEV infection after alloHSCT is easily overlooked and under-diagnosed and should be searched for systematically by NAT in cases of LFT abnormalities. Although the course of HEV infection was rather benign in this retrospective series, the burden of HEV among alloHSCT recipients needs to be investigated. Larger studies are needed to highlight factors associated with viral clearance, with or without treatment, in this highly immunosuppressed population.

## Figures and Tables

**Figure 1 viruses-11-00622-f001:**
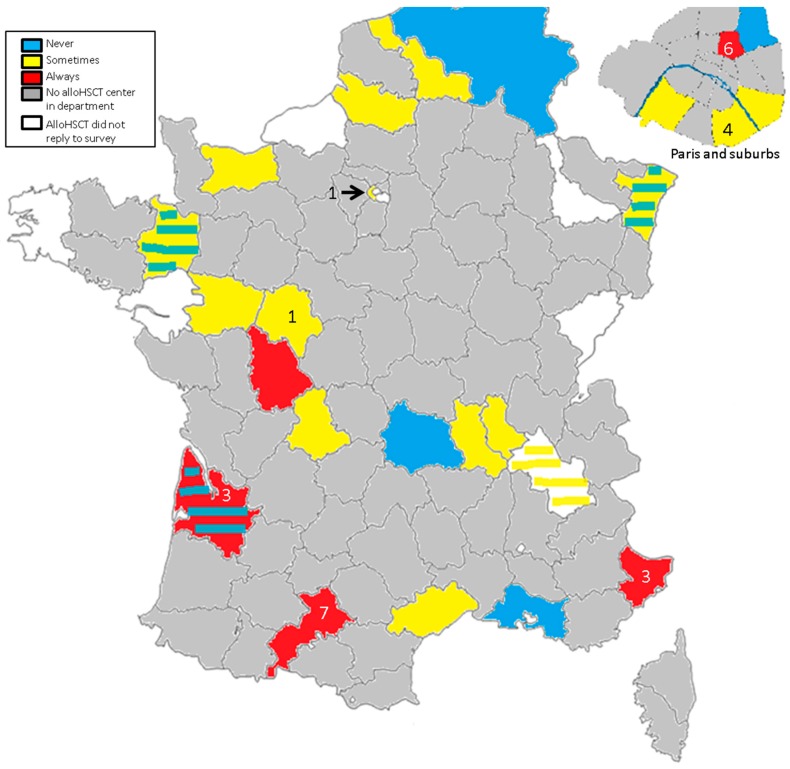
Awareness of centers of the burden of hepatitis E virus (HEV) and breakdown of cases. Color code describes the frequency of HEV search in cases of liver function tests abnormalities. Numbers indicate the number of cases diagnosed in each center. Stripes indicate the different diagnostic procedures for adult and pediatric centers in the same department.

**Table 1 viruses-11-00622-t001:** Patient characteristics at diagnosis.

Variable	Value	Diagnosis before Tx	Diagnosis after Tx
*n* = (%)	25	3	22
Age (year)			
Median (range)	49 (23–68)		
Male sex	18 (72)		
Underlying hematological disease			
Acute myeloid leukemia	10 (40)	2 (66)	8 (36)
Acute lymphoid leukemia	2 (8)		2 (9)
Lymphoma	7 (28)	1 (33)	6 (27)
Chronic lymphocytic leukemia	3 (12)		3 (14)
Myelodysplastic syndrome	1 (4)		1 (4.5)
Myeloproliferative neoplasm	1 (4)		1 (4.5)
Multiple myeloma	1 (4)		1 (4.5)
Conditioning			
Myelo-ablative	11 (44)	3 (100)	8 (36)
Reduced-intensity/sequential	14 (56)		14 (64)
Donor			
Matched related	12 (48)	3 (100)	9 (41)
Matched unrelated	8 (32)		8 (36)
Mismatched unrelated	5 (20)		5 (23)
Stem cell source			
Peripheral blood stem cells	18 (72)		18 (82)
Bone marrow	6 (24)	3 (100)	3 (14)
Cord blood	1 (4)		1 (4)
Chemotherapy		3 (100)	2 (9)
Immunosuppressive therapy		N/A	18 (82)
Calcineurin inhibitor			13 (59)
Steroids			7 (32)
Mycophenolate mofetil			2 (9)
Neutropenia	0		
Lymphopenia < 1000/mm^3^ (*n* = 23)	10 (39)	1 (33)	9 (43)
Symptoms			
None	19 (76)	2 (66)	17 (77)
Asthenia	4 (16)	1 (33)	3 (14)
Vomiting	1 (4)	1 (33)	
Pruritus	1 (4)		1 (4.5)
Headache	1 (4)		1 (4.5)
Fever	1 (4)	1 (33)	
Jaundice	1 (4)		1 (4.5)
Abnormal LFT at diagnosis (expressed as xULN)	25 (100)		
AST (*n* = 20)			3.2 (1.6–48)
ALT (*n* = 21)			5.5 (1.1–68)
GGT (*n* = 20)			5.4 (1.5–45)
ALP (*n* = 12)			2.1 (1.1–4.5)
Bilirubin (*n* = 8)			1.8 (1.3–27)
Positive blood NAT	25 (100)		
Genotype			
3c	2 (8)		2 (9)
3f	4 (16)		4 (18)
Unknown	19 (76)	3 (100)	16 (73)
Median time from diagnosis to Tx (range) (days)		43 (8–88)	
Median time from Tx to diagnosis (range) (days)			283 (26–4177)
Median time from diagnosis to treatment (range) (days)		N/A	18 (1–229)

Abbreviations: ALP: alkaline phosphatase; ALT: alanine aminotransferase; AST: aspartate aminotransferase; GGT: gamma glutamyl transferase; LFT: liver function test; N/A: not applicable; NAT: nucleic acid testing; Pat: patient; Tx: transplantation; ULN: upper limit of normal.

**Table 2 viruses-11-00622-t002:** Liver histology in four patients with biopsy.

	Pre-alloHSCT		Time from HEV Diagnosis to Biopsy	Histology
	Serology	NAT		
Pat. 6 *	Neg	Neg	14 d	Chronic hepatitis, signs of cirrhosis, significant inflammation, minor signs of steatosis
Pat. 14 **	Neg	Pos	≥30 d	Eosinophilic infiltrate, regenerating activity
			≥120 d	Partial disorganization of lobular architecture, no inflammation, no acute hepatitis, regenerating activity
Pat. 10	IgM neg	Neg	180 d	Acute hepatitis, no fibrosis
Pat. 11	Not done	Not done	210 d	Minimal lobular inflammation, steatosis, no fibrosis

Abbreviations: alloHSCT: allogeneic hematopoietic stem cell transplant; D: days; NAT: nucleic acid testing; Neg: negative; Pat.: patient; Pos: positive. * Acknowledged chronic alcohol consumption. ** Chronic HEV infection with positive NAT before alloHSCT, diagnosed one year after alloHSCT.

**Table 3 viruses-11-00622-t003:** Pattern of serology before and at HEV diagnosis.

Patient		Before alloHSCT	After alloHSCT but before HEV Diagnosis	At HEV Diagnosis
HEV infection preceding alloHSCT, diagnosed after alloHSCT				
13	IgG	Pos		Neg
	IgM	Pos		Pos
14	IgG	Neg		Neg
	IgM	Neg		Neg
24	IgG			Neg
	IgM			Neg
Acute HEV after alloHSCT				
1	IgG		Neg	Neg
	IgM		Neg	Pos
2	IgG			Pos
	IgM			Pos
3	IgG	Neg		Neg
	IgM	Neg		Pos
4	IgG			Pos
	IgM			Pos
5	IgG	Neg		
	IgM	Neg		
6	IgG	Neg		Pos
	IgM	Neg		Pos
7				
8	IgG			Neg
	IgM			Neg
9	IgG			Neg
	IgM			Neg
10				
11				
12	IgG			Pos
	IgM			Pos
15	IgG	Pos	Pos	
	IgM	Q	Neg	
16	IgG	Neg		Neg
	IgM	Neg		Neg
17	IgG	Neg		Pos
	IgM	Neg		Pos
21	IgG	Neg		Neg
	IgM	Neg		Neg
22				
23				
25	IgG			Neg
	IgM			Neg

Abbreviations: alloHSCT: allogeneic hematopoietic stem cell transplantation; Q: questionable; Neg: negative; Pos: positive. Blank spaces indicate no serology performed.

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
