# Peer review of "Hepatitis E and Allogeneic Hematopoietic Stem Cell Transplantation: A French Nationwide SFGM-TC Retrospective Study"

_viruses, 2019, doi:10.3390/v11070622_

Round 1
Reviewer 1 Report
The amendments has been done.
Author Response
We thank the reviewer
Reviewer 2 Report
The major limitation was and is the heterogenecity of the data, as described in the first review.
Author Response
We agree that this a limitation of our study, due to its retrospective and multicentric design
Reviewer 3 Report
Review report
Xhaard et al. HEV and allogeneic hematopoietic stem cell transplantation: a French nationwide SFGM-TC retrospective study
General comments
Study on HEV diagnostic practices and HEV infection aspects in French allo HSCT centers.
Clearly written paper.
M&M
Sound methods. How was HEV typed? Would be helpful to include this in the methods section
Overall conclusion
Interesting and valuable retrospective overview of HEV diagnostic practices and HEV infection aspects in French allo HSCT centers.
Author Response
We thank the reviewer for their review. Like NAT detection, both commercially available kits and in-house RT-PCR techniques were used for HEV genotyping. This information has been added to the manuscript (line 65)
Reviewer 4 Report
The authors have presented an interesting epidemiological study regarding HEV infection in allogenic HSCT patients in France with some interesting discussion on the testing, treatment (given the side effects commonly associated with ribavirin) and long term consequences of HEV in this patient group.
I have some minor comments/suggestions:
In the methods section, although it states that questionnaires were sent to transplant centres in 2014 and 2015, the time period that the reported data was collected over is not stated. I.e., were hospitals asked to declare all their HEV diagnoses, NAT methods etc and testing parameters from when to when? And did they report any changes in their testing protocols over that time period?
Line 84 states that 20 patients were identified as HEV positive but table 1 states 25 patients.
It is difficult to determine from the data given how prevalent HEV is in the alloHSCT patient population and if it is any higher compared to the general population. Would the authors be able to include this?
In table 1, it might be useful to include the data for the whole patient population for some parameters such as underlying haematological disease, sex, conditioning, source, IS therapy, development of cytopenias, etc., allowing any HEV risk factors to be identified. This, most likely, is especially useful for readers not familiar with this patient group.
In the supplementary tables would it be possible to add whether the patient was under IS regime and when to the data on individual patients (tables 1A and 1B)?
It is interesting that 2 patients seem to have recovered from HEV infection without seroconverting (one with ribavirin and one without). Would the authors consider commenting on this at all?
In line 228- should ‘evaluable’ perhaps be ‘evaluated’?
The authors mentioned that alloHSCT patients often have abnormal LFTs, was their anything distinct about the HEV induced LFTs compared to non-HEV induced LFTs that may help direct clinicians to consider HEV?
The authors state that the differences found here compared to the study by Versluis et al, is most likely due to differences IS regime. Do the others know that the IS therapies were different in these patients or is this a postulation?
I am curious- presumably there were undiagnosed or misdiagnosed cases of HEV in the centres that did not test for HEV. Do the authors know if there was an unusually high incidence of idiosyncratic hepatitis in these centre, DILI, or other likely misdiagnoses?
Author Response
Please see the attachment
The authors have presented an interesting epidemiological study regarding HEV infection in allogenic HSCT patients in France with some interesting discussion on the testing, treatment (given the side effects commonly associated with ribavirin) and long term consequences of HEV in this patient group.
I have some minor comments/suggestions:
In the methods section, although it states that questionnaires were sent to transplant centres in 2014 and 2015, the time period that the reported data was collected over is not stated. I.e., were hospitals asked to declare all their HEV diagnoses, NAT methods etc and testing parameters from when to when? And did they report any changes in their testing protocols over that time period?
Questionnaires sent in 2014 and 2015 asked for the date of first HEV search in each center, knowing that most of these centers were not aware of HEV infection before 2010. Participating centers declared all HEV cases from date of first search in their center until 2014/2015. As stated in the Results section (lines 72 to 74), three cases were diagnosed between 2005 and 2009 in the French center that has worked the most on HEV infection in immunocompromised patients (alloHSCT and solid-organ transplant recipients), and the 22 remaining cases were diagnosed between 2010 and 2015. No center reported any changes in testing protocols over time.
Line 84 states that 20 patients were identified as HEV positive but table 1 states 25 patients.
The other 5 cases were diagnosed in two of the sixteen centers that occasionally searched for HEV if transaminases increased (lines 82-83).
It is difficult to determine from the data given how prevalent HEV is in the alloHSCT patient population and if it is any higher compared to the general population. Would the authors be able to include this?
Indeed, it would have been very interesting to report an incidence of HEV infection among alloHSCT recipients. Unfortunately, this is not possible due to the retrospective nature of our study.
As the HEV diagnostic period in our study (2010-2015) is not the same than the transplantation period (2000-2014), it is difficult to calculate the percentage of patients with HEV infection in relationship to the number of alloHSCT performed during the same time period.
In table 1, it might be useful to include the data for the whole patient population for some parameters such as underlying haematological disease, sex, conditioning, source, IS therapy, development of cytopenias, etc., allowing any HEV risk factors to be identified. This, most likely, is especially useful for readers not familiar with this patient group.
Table 1 gives these data for our entire population (n=25). Unfortunately, due to the retrospective design of our study, and because the HEV diagnostic period in our study (2010-2015) is not the same than the transplantation period (2000-2014) of our patients with HEV infection, we were not able to compare our population with the entire population of patients receiving an allogeneic HSCT in France. The aims of our study were to describe HEV diagnostic practices in alloHSCT centers and the course of HEV infection before or after alloHSCT, but not to identify HEV risk factors.
In the supplementary tables would it be possible to add whether the patient was under IS regime and when to the data on individual patients (tables 1A and 1B)?
Unfortunately, data on immunosuppressive regimen were only available at HEV diagnosis, and not during follow-up.
It is interesting that 2 patients seem to have recovered from HEV infection without seroconverting (one with ribavirin and one without). Would the authors consider commenting on this at all?
As indicated in the Discussion section (lines 193-194), aberrant serodynamics have previously been reported among alloHSCT recipients (Versluis et al, Blood 2013), and are usually explained by the impaired immune reconstitution after allogeneic HSCT. This information has been added to the manuscript (line 194).
In line 228- should ‘evaluable’ perhaps be ‘evaluated’?
We confirm ‘evaluable’. Patient 14 (line 160) died during ribavirin treatment, with decreasing but still positive NAT, and was therefore not available for the evaluation of ribavirin efficacy, leaving us with 12 evaluable patients out of 13 treated patients.
The authors mentioned that alloHSCT patients often have abnormal LFTs, was their anything distinct about the HEV induced LFTs compared to non-HEV induced LFTs that may help direct clinicians to consider HEV?
Unfortunately, the retrospective design of our study does not allow us to compare HEV-induced LFTs with non-HEV induced LFTs. That comparison could only be performed in a prospective study, with systematic HEV search in case of LFTs abnormalities after allogeneic HSCT.
The authors state that the differences found here compared to the study by Versluis et al, is most likely due to differences IS regime. Do the others know that the IS therapies were different in these patients or is this a postulation?
In the article by Versluis et al, data on immunosuppressive treatment at the time of HEV diagnosis only indicate which immunosuppressive medications were given, but at which doses. Moreover, no systematic evaluation of immune system (gammaglobulins, immunophenotyping, etc…) was performed in either study at the time of HEV diagnosis, so this is a postulation.
I am curious- presumably there were undiagnosed or misdiagnosed cases of HEV in the centres that did not test for HEV. Do the authors know if there was an unusually high incidence of idiosyncratic hepatitis in these centre, DILI, or other likely misdiagnoses?
Unfortunately, this information is not available due to the retrospective nature of our study.
